# Mechanochemical ligand-controlled regiodivergent hydroarylation of alkenes via iron-catalyzed C−H activation

Zi-Jing Zhang[1,2], Ziyue Liu[1,2], Xinran Chen[1,2] & Lutz Ackermann [1] ✉

The iron-catalyzed hydroarylation of alkenes with indoles is a sustainable, effective synthetic transformation towards the construction of functionalized indoles - crucial motifs for various bioactive molecules and drug candidates. However, such transformations have proven challenging for unactivated alkenes, and the requirement for (super)stoichiometric amounts of reactive Grignard reagents has limited broader applications. Herein, we address these major challenges by integrating iron/N-heterocyclic carbene-catalyzed C−H activation with mechanochemistry techniques. This approach enables mechanochemical iron-catalyzed anti-Markovnikov hydroarylations of unactivated alkenes using a bis(N-heterocyclic carbene) ligand, as well as regiodivergent hydroarylation of aryl alkenes by varying the N-heterocyclic carbene ligand. To this end, magnesium metal serves as a convenient reductant to form catalytically active iron(0) species by mechanochemistry, thereby improving the sustainability and functional group compatibility. Experimental and computational studies elucidate the possible catalytic mode of action, and a data science analysis captured the key features of the N-heterocyclic carbene ligands in controlling regiodivergent selectivity.

Over the past decade, mechanochemically promoted organic transformations have gained major momentum due to beneficial features, including shorter reaction times, lower catalyst loadings, unique reactivity, and most notably the reduction of harmful, often toxic organic solvents[1–4]. In this context, pioneering studies by Bolm[5–9], Ito[10,11], and others[12–15] have demonstrated the potential of mechanochemistry in enabling bond activations with precious transition metal catalysts, thus reflecting its unique potential for sustainable molecular assembly.

C−H alkylation via the direct regioselective addition of indoles onto alkenes is arguably the most efficient and atom-economical approach to obtain functionalized indoles[16,17], which are omnipresent structural motifs in drug development due to their strong affinity for biological targets (Fig. 1a)[18,19]. Despite indisputable advances, these hydroarylations largely relied on the use of toxic, cost-intensive and precious metal catalysts[20,21], while transformations involving more sustainable 3d metal catalysts remain limited[22–31]. Recently, Yoshikai[32]

and Ackermann[33–35] groups developed C−H alkylation of indoles and styrenes using the most abundant, low-cost, and non-toxic iron catalyst[36–41] with monodentate N-heterocyclic carbene (NHC) ligands. However, several inherent limitations in these approaches still persist and need to be addressed. The use of (super)stoichiometric amounts of Grignard reagents drastically reduces the reaction sustainability as well as their functional group tolerance[42–44]. In addition, the reported methods have only achieved Markovnikov-selective hydroarylation of aryl alkenes[32–35] or anti-Markovnikov-selective hydroarylation of heteroatom-substituted alkenes[35,40,41], while transformations involving unactivated alkenes remain challenging.

The Grignard reagent serves a pivotal role in iron/NHC catalysis, and, as a strong reductant, is responsible for the generation of catalytically active low-valent iron species[32–34]. The in situ generation of Grignard reagent from aryl halides and magnesium metal through the mechanochemical technique has been achieved to avoid the sensitive

[1]Wöhler Research Institute for Sustainable Chemistry (WISCh), Georg-August-Universität Göttingen, Göttingen, Germany. [2]These authors contributed equally: Zi-Jing Zhang, Ziyue Liu, Xinran Chen. ✉e-mail: Lutz.Ackermann@chemie.uni-goettingen.de

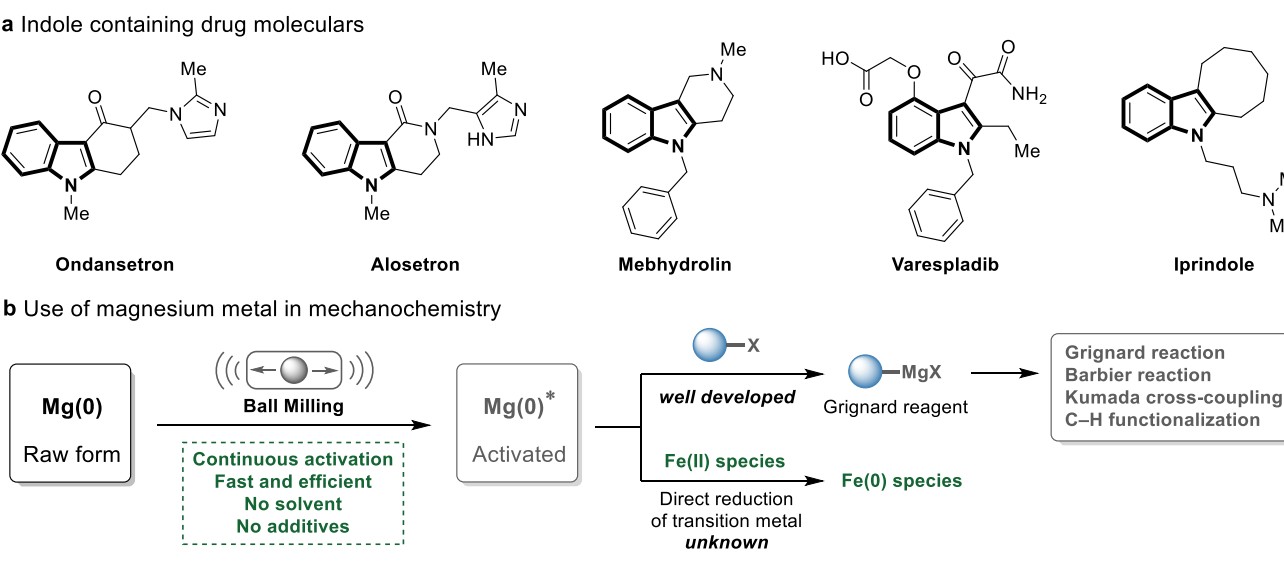

**a** Indole containing drug molecolars

Ondansetron    Alosetron    Mebhydrolin    Varespladib    Iprindole

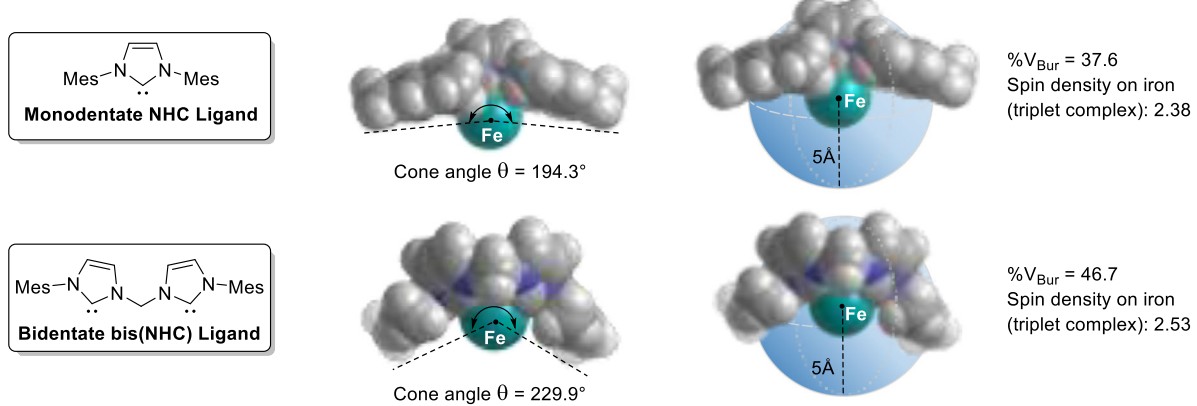

**b** Use of magnesium metal in mechanochemistry

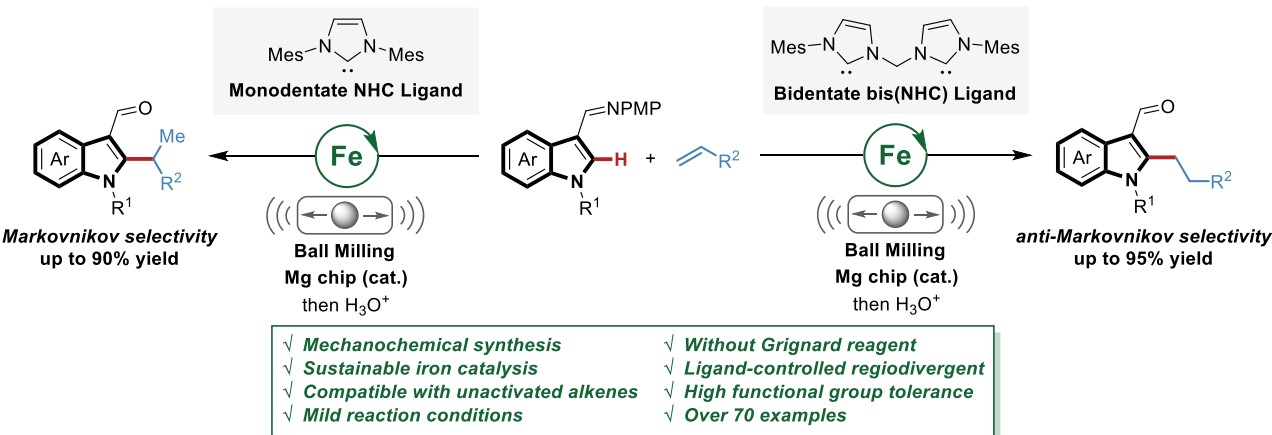

**c** Key features comparison of representative iron(0)-NHC complexes

Monodentate NHC Ligand

Cone angle θ = 194.3°

%V_Bur = 37.6
Spin density on iron
(triplet complex): 2.38

Bidentate bis(NHC) Ligand

Cone angle θ = 229.9°

%V_Bur = 46.7
Spin density on iron
(triplet complex): 2.53

**d** This work: mechanochemical iron-catalyzed regiodivergent hydroarylation

Monodentate NHC Ligand

*Markovnikov selectivity*
up to 90% yield

Fe
Ball Milling
Mg chip (cat.)
then $H_3O^+$

Bidentate bis(NHC) Ligand

Fe
Ball Milling
Mg chip (cat.)
then $H_3O^+$

*anti-Markovnikov selectivity*
up to 95% yield

√ *Mechanochemical synthesis*          √ *Without Grignard reagent*
√ *Sustainable iron catalysis*          √ *Ligand-controlled regiodivergent*
√ *Compatible with unactivated alkenes*  √ *High functional group tolerance*
√ *Mild reaction conditions*            √ *Over 70 examples*

**Fig. 1 | Towards new sustainable assembly of functionalized indoles by mechnochemical iron-catalyzed regiodivergent hydroarylation. a** Indole-containing drug molecules. **b** Use of magnesium metal in mechanochemistry. **c** Key features comparison of representative iron(0)-NHC complexes. **d** This work: mechanochemical iron-catalyzed regiodivergent hydroarylation.

and tedious preparation steps (Fig. 1b)[45–51]. This one-pot approach benefits from the capability of mechanochemistry for the direct activation of zero-valent magnesium[52–54] by an electron transfer process. Besides, magnesium metal has proven to be a useful alternative to Grignard reagents in cobalt-catalyzed hydroarylation, albeit with high reaction temperature and narrower scopes[55]. However, the direct use of magnesium metal as a reductant in iron-catalyzed hydroarylation has thus far proven elusive. In this context, we inferred that magnesium metal might solely serve as a reductant under mechanochemical conditions, circumventing the involvement of the Grignard reagent, to generate a catalytically active low-valent iron species for the hydroarylation.

**Fig. 2 | Screening of NHC preligands for mechanochemical iron-catalyzed hydroarylation of unactivated alkenes.** Reaction conditions: **1a** (0.1 mmol), **2a** (0.15 mmol), Fe(acac)$_3$ (10 mol%), **NHC preligand** (10 mol%), Mg chip (50 mol%), TMEDA (0.1 mmol) and THF (50 μL) were placed in a stainless-steel vessel (5 mL) with stainless-steel ball ($d_{MB}$ = 7 mm), milled in a mixer mill (RETSCH MM 400) at 30 Hz for 180 min under nitrogen atmosphere. Then THF (3 mL) and HCl aq. (3 M, 1 mL) were added, and the mixture was stirred for 2 h. The ratio of l:b is > 99:1 for all cases. The yield was determined by $^1$H NMR spectroscopy using 1,3,5-trimethoxybenzene as the internal standard. $^a$In the absence of TMEDA: 72% yield; in the absence of THF: 83% yield; in the absence of Fe(acac)$_3$, **NHC preligand**, or Mg chip: n.d. TMEDA, *N,N,N',N'*-tetramethylethylenediamine; THF, tetrahydrofuran; Mes, 2,4,6-trimethylphenyl; Dipp, 2,6-diisopropylphenyl; Xyl, 2,6-dimethylphenyl; n.d., not detected.

NHCs represent an indispensable class of ligands for transition-metal catalyzed C−H activations[56,57]. Their strong σ-donating ability contributes to the more stable metal–ligand bond, thereby enhancing the robustness of the metal complexes[58−60]. Compared with monodentate NHC ligands, bidentate bis(NHC) ligands exhibit distinctive spatial characters and electronic effect[61,62]. Taking low-valent iron(0) complex as an example, bis(NHC) complex has a larger cone angle[63] and buried volume[64], which potentially brings opportunities for precise regioselectivity control (Fig. 1c). Moreover, the location of higher spin density at the iron center in bis(NHC) complex may lead to underdeveloped reactivity[65,66], presenting avenues for further exploration.

Herein, aiming at addressing the limitations of iron/NHC-catalyzed C−H functionalization, we develop a mechanochemical iron-catalyzed regiodivergent hydroarylation of alkenes with indoles via C−H activation (Fig. 1d). The salient features of this strategy include: (1) the use of magnesium metal as a convenient reductant in iron-catalyzed hydroarylation with the aid of mechanochemistry, which improves the sustainability and functional group compatibility; (2) *anti*-Markovnikov-selective hydroarylation of unactivated alkenes using an iron/bis(NHC) catalytic system; and (3) regiodivergent hydroarylation of aryl alkenes by the judicious choice of the NHC ligands. Further detailed experimental and theoretical mechanistic studies allow shedding light on the iron/NHC catalytic system from a broader perspective.

## Results and discussion
### Optimization of reaction conditions
We initiated our studies into the hydroarylation of 1-octene **2a** with indole derivative **1a** using the iron/NHC catalytic system under mechanochemical conditions (Fig. 2). We probed the reaction using different types of monodentate NHC preligands (**L1**−**L7**), which were unable to deliver the desired product. Subsequently, we attempted the reaction using bis(NHC) preligands. Pleasingly, the desired alkylated indole **3** was obtained in 21% yield with *anti*-Markovnikov selectivity when using Fe(acac)$_3$ as the metal catalyst, **L8** as the preligand and magnesium chip as the reductant under mechanochemical conditions. Thereafter, the structure of bis(NHC) preligand was further explored. A drastic increase was observed upon the substitution of the methyl group in **L8** by the bulkier isopropyl (**L9**), where the alkylated indole was obtained in 84% yield. Further improvement in the yield (92%) was achieved by using 2,4,6-trimethylphenyl as the substituent (**L10**). When applying phenyl or 2,6-diisopropylphenyl as the substituent (**L11** and **L12**), only trace amounts of the alkylated product were obtained, indicative of the relevance of stereoelectronic features. The use of a bis(NHC) preligand with saturated backbones (**L13**) and a tridentate NHC preligand (**L15**) resulted in low reaction efficiency, while a bis(NHC) preligand bridged by two methylene groups (**L14**) completely inhibited reactivity. Finally, the reaction conditions were further optimized (see *Supplementary Information* Table S2 and S3 for details). The reaction efficiently proceeded even under additive-free or solvent-free conditions, albeit with a slight drop in yield. Control experiments indicated that the iron catalyst, the bis(NHC) preligand, and the magnesium chip were crucial for the desired transformation. Notably, replacing ball milling by thermal reaction conditions completely failed to furnish the desired product, even at an elevated reaction temperature of 100 °C.

### Substrate scope
With the optimal bis(NHC) preligand (**L10**) in hand, we evaluated the viable substrate scope for hydroarylation of unactivated alkenes under the optimized reaction conditions to delineate the full potential and

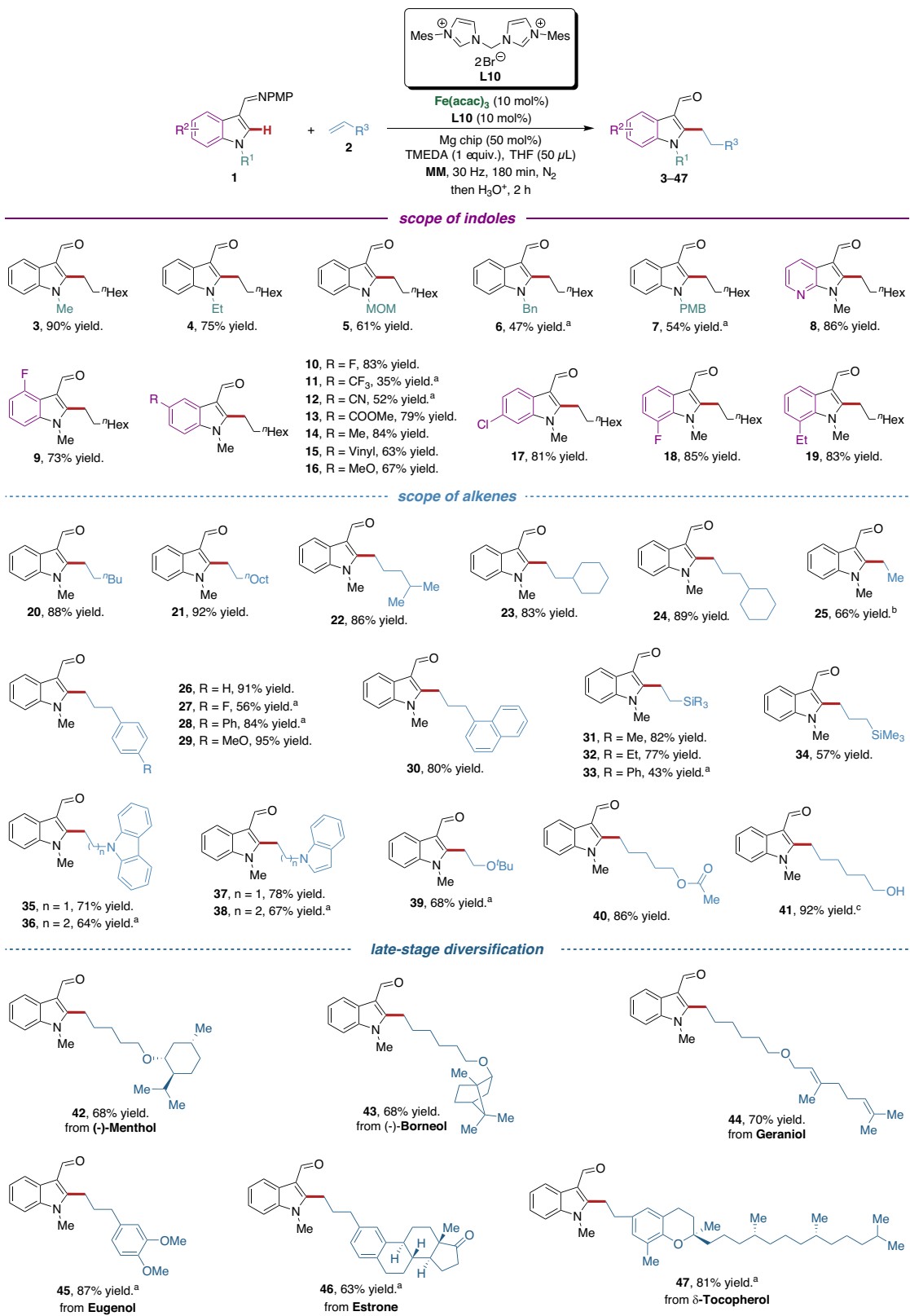

**Fig. 3 | Substrate scope for mechanochemical iron-catalyzed hydroarylation of unactivated alkenes.** Reaction conditions: **1** (0.2 mmol), **2** (0.3 mmol), Fe(acac)₃ (10 mol%), **L10** (10 mol%), Mg chip (50 mol%), TMEDA (0.2 mmol) and THF (50 μL) were placed in a stainless-steel vessel (5 mL) with stainless-steel ball ($d_{MB}$ = 7 mm), milled in a mixer mill (RETSCH MM 400) at 30 Hz for 180 min under nitrogen atmosphere. Then THF (3 mL) and HCl aq. (3 M, 1 mL) were added and the mixture was stirred for 2 h. The ratio of l:b is >99:1 for all cases. Yields are those of the isolated products. [a]270 min. [b]Using ethoxyethene as a substrate. [c]Using *tert*-butyl(hex-5-en-1-yloxy)dimethylsilane as substrate with the reaction quenched by HCl aq. (3 M, 1 mL).

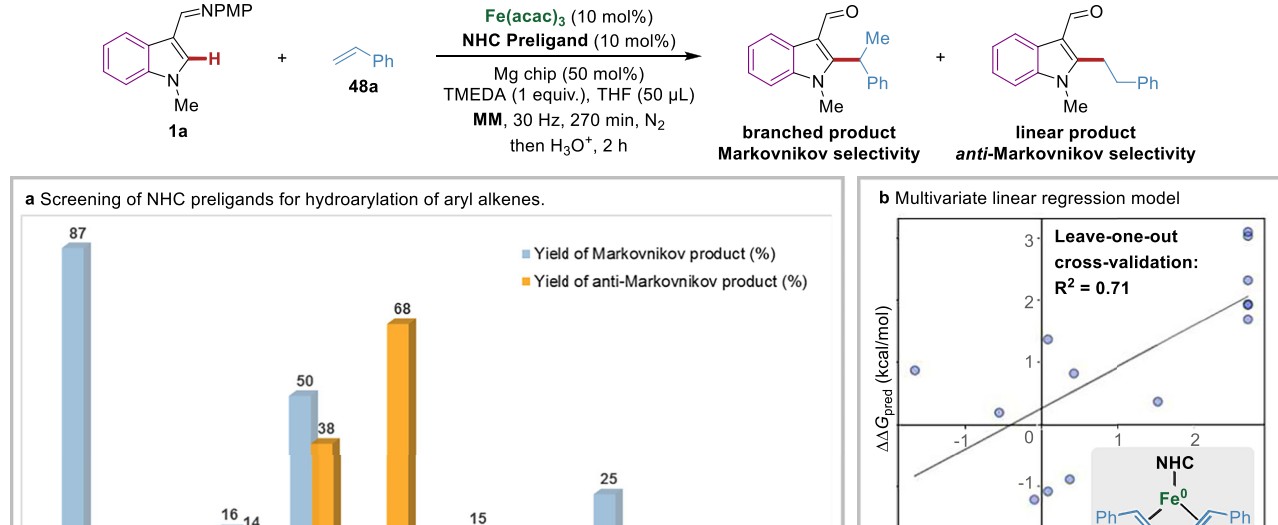

**Fig. 4 | Optimization of catalyst systems for mechanochemical iron-catalyzed hydroarylation of aryl alkenes. a** Screening of NHC preligands for hydroarylation of styrene. **b** Preliminary multivariate linear regression model for the structure-regioselectivity relationship in iron-catalyzed hydroarylation of styrene. Four parameters for key iron(0)-NHC complexes were involved, including cone angle θ, percent buried volume %$V_{Bur}$ at sphere radius of 5.0 Å and 3.5 Å and spin density on iron center.

robustness of this transformation (Fig. 3). Indoles bearing a variety of *N*-substituents gave the corresponding products **3–7** in moderate to good yields. The azaindole proved to be a suitable substrate, affording alkylated indole **8** in 86% yield. As expected, the presence of either electron-withdrawing or electron-donating substituents at 4-, 5-, 6- or 7-position of the indole ring was well tolerated, delivering the desired products **9–19** in high yields. Notably, cyano and ester substituents, which are incompatible under the Grignard conditions, were well tolerated in our mechanochemical approach (**12,13**). In addition, the reaction conditions turned out to be generally compatible with various unactivated alkenes. An array of linear and branched aliphatic alkenes revealed to be efficient in the reaction, providing the desired products **20–24** in excellent yields. When employing ethoxyethene as the substrate, ethyl-substituted indole **25** was obtained in 66% yield. A variety of allylbenzenes containing either electron-withdrawing or electron-donating substituents performed well, providing the corresponding products **26–30** in high yields. Silicon-, nitrogen- and oxygen-containing alkenes effectively engaged in the reaction, furnishing the corresponding products **31–41** in good yields. Noteworthy, this hydroarylation could be further applied for the late-stage modification of complex molecules. As displayed, (-)-menthol, (-)-borneol, geraniol, eugenol, estrone and δ-tocopherol-conjugated indoles **42–47** could be easily accessed in high yields. Remarkably, complete *anti*-Markovnikov selectivity was observed in all cases.

Next, we explored the effects of NHC ligands in the hydroarylation of aryl alkenes (Fig. 4). Using styrene **48a** as the model substrate, the mechanochemical iron-catalyzed hydroarylation was first investigated by monodentate NHC preligands (Fig. 4a). Here, several monodentate NHC preligands (**L1 – L6**) all produced completely Markovnikov products (b:l > 99:1), among which **L1** gave the highest yield (87%), whereas no product was obtained when using **L7**. A series of bis(NHC) preligands were then screened, revealing that **L10** was the most suitable one for *anti*-Markovnikov selectivity (b:l = 6:94), while others, such as **L8, L9, L11 – L15** displayed low efficacy and poor regioselectivity. These diverse selectivity data of styrene with different ligands encouraged us to apply the multivariate linear regression (MVLR)

approach[67] for capturing the structure-regioselectivity relationship in mechanochemical iron-catalyzed hydroarylation. A preliminary MVLR model was developed using four physical organic parameters of the key triplet iron(0)-NHC complexes (Fig. 4b), including cone angle θ, percent buried volume %$V_{Bur}$ (sphere radius at 5.0 and 3.5 Å) and spin density on the iron center (*vide supra*). These four parameters were able to provide reasonable prediction on the regioselectivity of hydroarylation using aryl alkenes with an $R^2$ value of 0.71 in leave-one-out cross-validation. Among them, the cone angle has the largest coefficient in the normalized regression equation, highlighting the significance of ligand geometry in determining regioselectivity. Nevertheless, the steric environment around the iron center and the electron-donating ability of NHC also contribute to the regioselectivity control.

Having established the optimized reaction conditions, we turned to evaluate the substrate scope for this ligand-controlled regiodivergent hydroarylation of aryl alkenes (Fig. 5). First, the scope of Markovnikov-selective hydroarylation catalyzed by the monodentate preligand **L1** was tested. Indoles bearing cyano, ester, or chloro groups on the aromatic ring were well-tolerated, resulting in the formation of alkylation products **49–55** in good yields with excellent regioselectivities (b:l > 99:1 for all). A wide range of styrenes as well as vinylferrocene were competent substrates in the Markovnikov-selective hydroarylation, giving products **56–63** in high yields (b:l > 99:1 for all). Subsequently, we examined the scope of *anti*-Markovnikov-selective hydroarylation catalyzed by the bidentate preligand **L10**. The substitution pattern and electronic properties of the substituents at 5- or 6- position of the aromatic ring did not affect the reaction efficiency and regioselectivity, giving rise to the corresponding products **64 – 69** in good yields with high levels of regioselectivity (l:b = 92:8 to 97:3). While, the substituent at the 7-position of indole, which is spatially proximal to the 2-position, imposed steric hindrance that affected the bond formation at the 2-position (**70**). Styrenes bearing electron-withdrawing or electron-donating substituents on the *para-*, *meta-* or *ortho-*position of aromatic ring, reacted well to give the desired products **71 – 77** in 51–86% yields (l:b = 85:15 to 98:2). Remarkably,

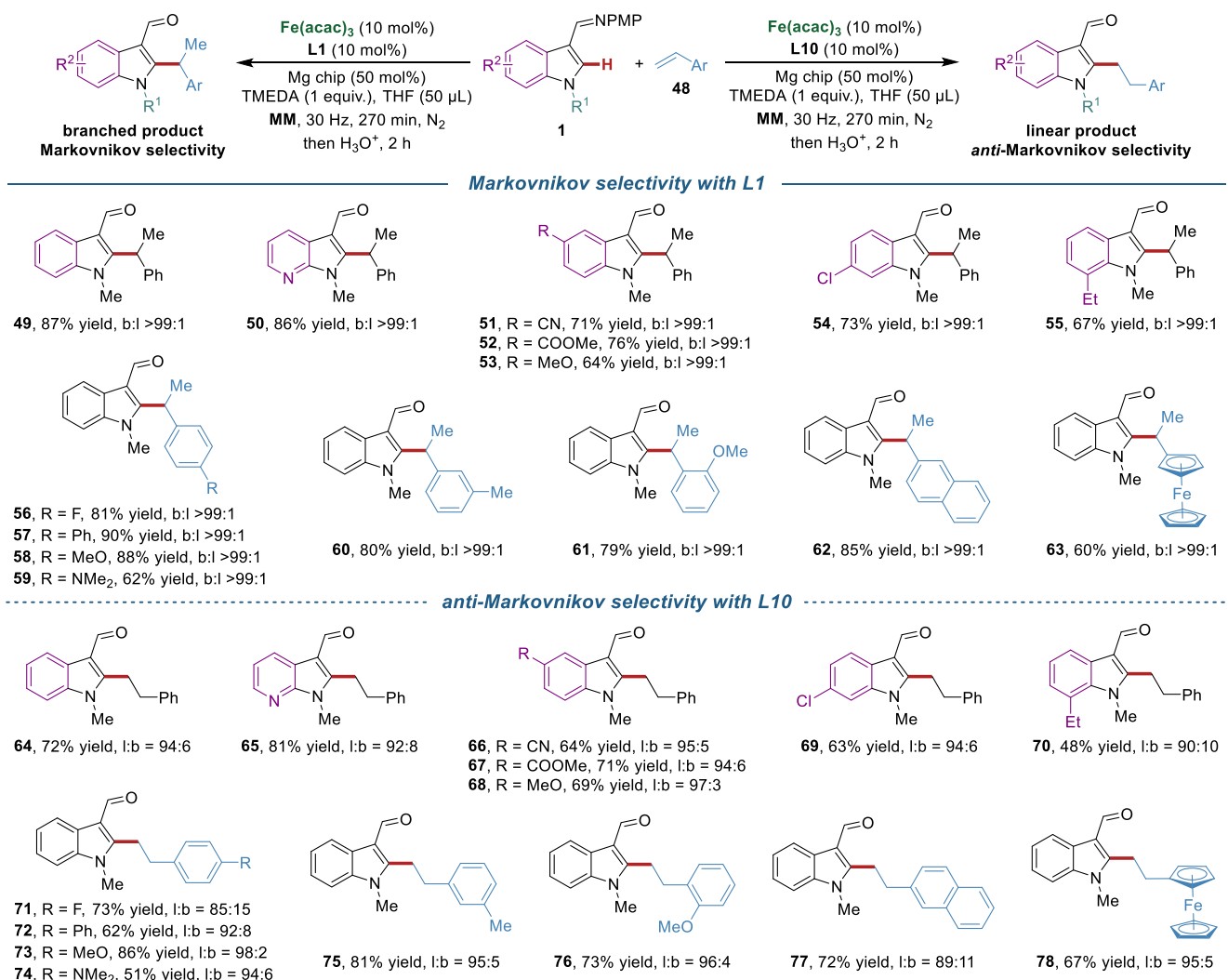

**Fig. 5 | Substrate scope for ligand-controlled regiodivergent hydroarylation of aryl alkenes.** Reaction conditions: **1** (0.2 mmol), **48** (0.3 mmol), Fe(acac)₃ (10 mol %), **NHC preligand** (10 mol%), Mg chip (50 mol%), TMEDA (0.2 mmol) and THF (50 μL) were placed in a stainless-steel vessel (5 mL) with stainless-steel ball ($d_{MB}$ = 7 mm), milled in a mixer mill (RETSCH MM 400) at 30 Hz for 270 min under nitrogen atmosphere. Then THF (3 mL) and HCl aq. (3 M, 1 mL) were added, and the mixture was stirred for 2 h. Yields are those of the isolated products.

vinylferrocene was also compatible with the optimal reaction conditions, affording the desired product **78** in good yield (67%) and regioselectivity (l:b = 95:5).

## Scale-up and late-stage transformations

To further demonstrate the utility of this method, a scale-up reaction and several late-stage transformations of alkylated indole products were carried out (Fig. 6). The current strategy was shown to be highly robust and scalable, as a gram-scale reaction of **1a** and **2a** proceeded smoothly, generating **3** in 86% yield (Fig. 6a). The alkylated indole-3-carboxaldehyde products act as synthetically useful intermediates that can be readily transformed into complex architectures using standard methods. The presence of the formyl group enables extensive structural diversification through subsequent functionalization. It could be removed under palladium catalysis, yielding indole **79** (Fig. 6b). The treatment with Grignard reagent delivered the corresponding alcohol **80** (Fig. 6c). A Wittig reaction produced the internal olefin **81** (Fig. 6d). Reductive amination using morpholine efficiently furnished the corresponding amine **82** (Fig. 6e). Furthermore, the formyl group was also able to be a linchpin for condensation, enabling the access to barbiturate-decorated indole **83** (Fig. 6f). Subsequently, we employed

the formyl group as a weakly coordinating directing group to facilitate C4−H activation. Thereby, C4-functionalized indoles **84** − **86** could be synthesized in good yields (Fig. 6g−i). Ultimately, intramolecular crotonization of **86** led to 3,4-fused tricyclic indole **87** (Fig. 6j).

## Mechanistic studies

To gain mechanistic insights into the modus operandi of the regioselective hydroarylation of unactivated alkenes, experiments and density functional theory (DFT) studies were carried out (Fig. 7). We synthesized a well-defined iron(II)-bis(NHC) complex[65] from preligand **L10** and demonstrated its catalytic activity under optimal conditions (Fig. 7a). However, it failed to catalyze the reaction in the absence of magnesium chip, indicating that the zero-valent iron-bis(NHC) complex produced by the mechanochemical magnesium reduction might be the active species. The reaction of the C2-deuterated indole [D]₁-**1a** with 1-octene **2a** afforded the product [D]₁-**3** (Fig. 7b). The transfer of the C2-deuterium from indole to two methylene groups implied the reversibility of the alkene insertion step. A deuterium-scrambling experiment was then performed using one equivalent of a mixture of substrates [D]₁-**1a** and **1n** under standard reaction conditions (Fig. 7c). H/D crossover observed in both products suggested that the

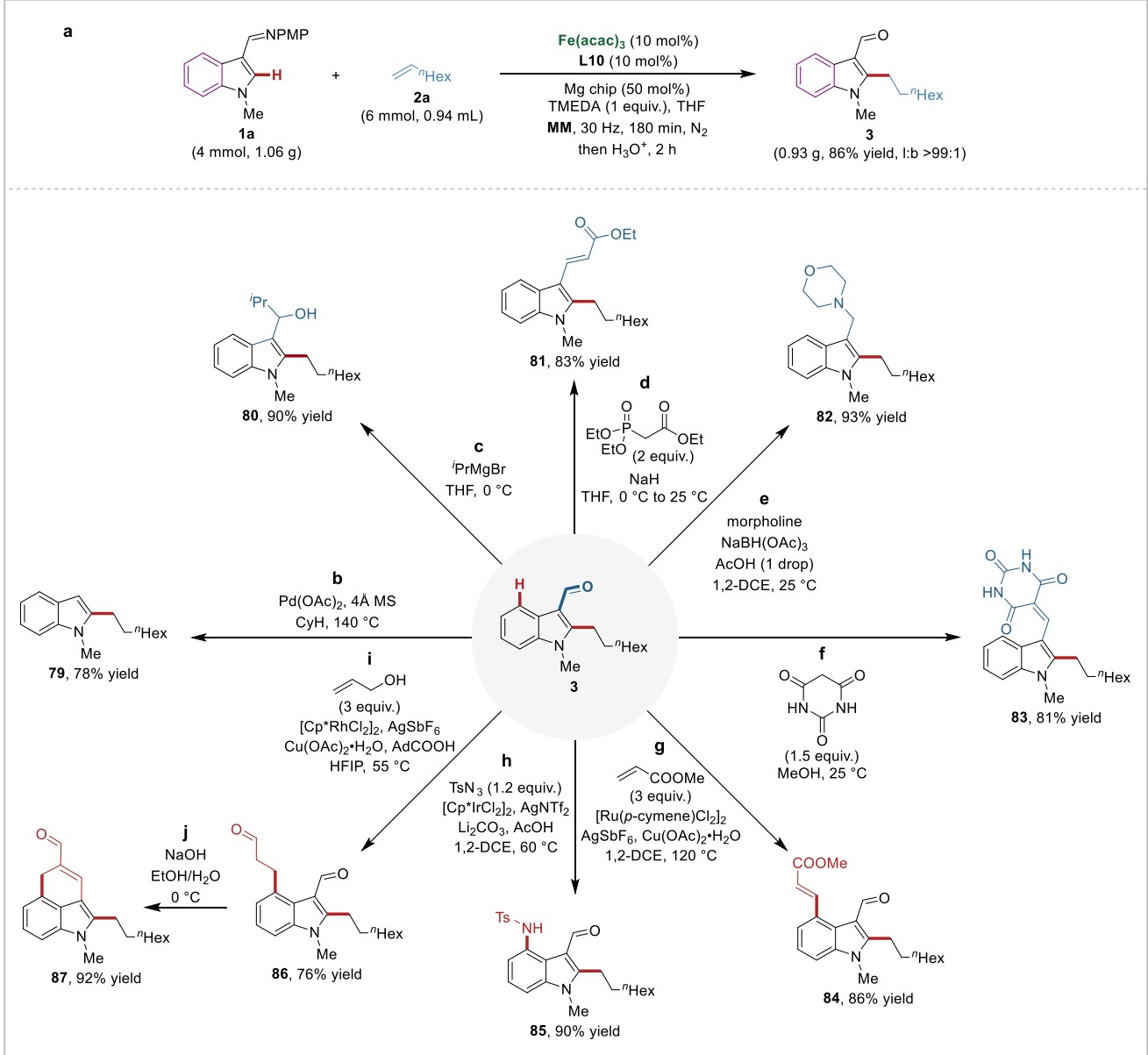

**Fig. 6 | Scale-up reaction and diversification of alkylated indole products.**
**a** Gram-scale reaction of **1a** and **2a**. **b** Transformation to indole **79** under palladium catalysis. **c** Transformation to alcohol **80**. **d** Transformation to olefin **81**.
**e** Transformation to amine **82**. **f** Transformation to barbiturate decorated indole **83**. **g** C4-alkenylation of alkylated indole product **3** under ruthenium catalysis. **h** C4-amination of alkylated indole product **3** under iridium catalysis. **i** C4-alkylation of alkylated indole product **3** under rhodium catalysis. **j** Intramolecular crotonization of **86** to afford 3,4-fused tricyclic indole **87**.

dissociation and re-coordination of the alkene before the formation of alkyl iron species was possibly a reversible process.

## Computational studies

Based on the experimental results, DFT calculations were conducted to investigate the reaction mechanism and the origins of regioselectivity[68], using indole **1a** and 1-octane **2a** as model substrates. It is worth noticing that the solvent effect was included in the free energy calculation based on gas-phase optimized geometries, given that our reaction is liquid-assisted grinding with THF[69]. The free energy diagram of the most favorable pathway for iron/NHC-catalyzed hydroarylation starting from iron(0) complex **int1** is shown in Fig. 7d and *Supplementary Information* Supplementary Fig. S11. As the coordination of the substrate **1a** requires a barrier of 23.9 kcal/mol via transition state **TS4**, it is most likely to be the rate-determining step in the catalytic cycle. The subsequent C–H activation occurs

through an oxidative addition of the C–H bond onto the iron center, generating iron hydride species **int7**. The calculated alkene insertion is facile and reversible, consistent with deuterium-labeling experiments. Then, the resulting alkyl-iron(II) species **int9** undergoes reductive elimination forming C–C bond, which is probably the regioselectivity-determining step. Two competing reductive elimination transition states, **TS10-linear** and **TS10-branched,** generating the respective linear and branched products are presented in Fig. 7e. For unactivated alkene **2a**, **TS10-linear** is 4.7 kcal/mol more favorable than **TS10-branched** in terms of free energy. In **TS10-linear**, the primary alkyl is positioned away from the substrate, allowing a favorable π-π stacking interaction between PMP and mesityl groups. In contrast, the secondary alkyl in **TS10-branched** is more steric-demanding, leading to repulsion between indole and alkyl chain, thereby increasing the reaction barrier for branched product. With the combination of experimental and computational

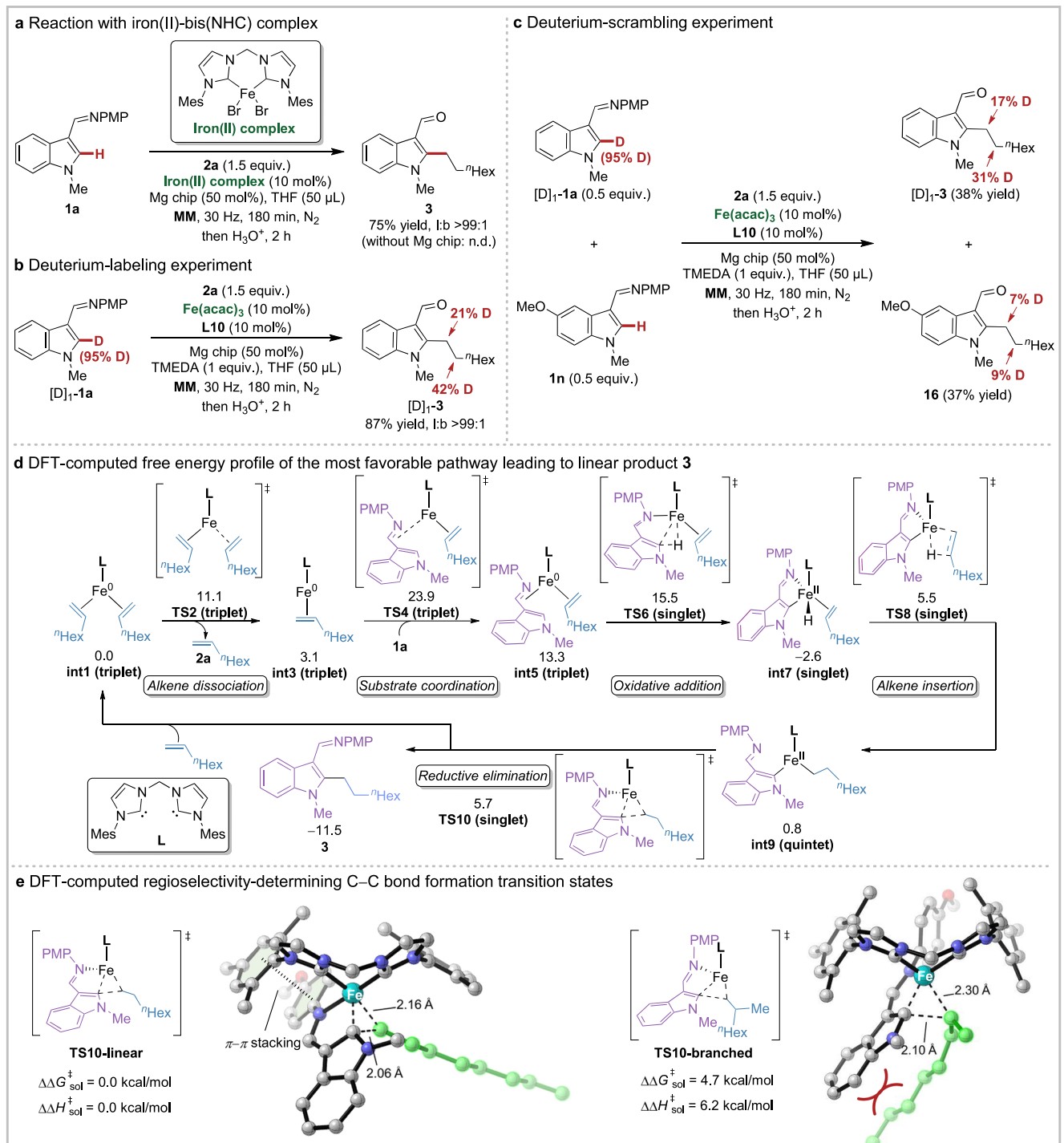

**Fig. 7 | Mechanistic studies. a** Reaction with iron(II)-bis(NHC) complex.
**b** Deuterium-labeling experiment. **c** Deuterium-scrambling experiment.
**d** Mechanism for the generation of linear product **3**. Computational method at

TPSSh-D3(BJ)/def2-TZVPP-SMD(Tetrahydrofuran)//B3LYP-D3(BJ)/def2-SVP level of theory. **e** DFT-computed regioselectivity-determining reductive elimination transition states forming C–C bond of linear and branched products.

studies, the reaction mechanism of iron-catalyzed hydroarylation with bis(NHC) ligand was clarified, along with the rationale of the regioselectivity of representative unactivated alkene.

We have successfully developed a sustainable and highly effective ligand-controlled regiodivergent hydroarylation of alkenes via mechanochemical iron-catalyzed C−H activation, leading to functionalized indoles with high structural diversity. The *anti*-Markovnikov-selective hydroarylation of unactivated alkenes could be achieved by the aid of a bis(NHC) ligand, while the regiodivergent hydroarylation of

aryl alkenes was accomplished by modifying the structure of NHC ligands. Magnesium metal serves as a convenient reductant to form the catalytically active iron(0) species under mechanochemical conditions, making the reaction sustainable, environmentally friendly, and compatible with a broad range of functional groups. Moreover, detailed mechanistic studies and data science analysis revealed the reaction mechanism and elucidated the key features of NHC ligands in controlling the regioselectivity, setting the stage for the further application of this iron/NHC catalytic system.

## Methods

### General procedure for mechanochemical iron-catalyzed regioselective hydroarylation of unactivated alkenes

In the glove box, a mixture of indole substrate **1** (0.2 mmol), unactivated alkene **2** (0.3 mmol), Fe(acac)$_3$ (10 mol%, 0.02 mmol, 7.1 mg), **L10** (10 mol%, 0.02 mmol, 10.9 mg), magnesium chip (50 mol%, 0.1 mmol, 2.4 mg), TMEDA (0.2 mmol, 30 μL) and tetrahydrofuran (50 μL) were placed in a nitrogen-purged stainless-steel vessel (5 mL) with a stainless-steel ball ($d_{MB}$ = 7 mm). Then, the vessel was sealed and milled in a mixer mill (RETSCH MM 400) at 30 Hz for 180 min under a nitrogen atmosphere. Then, the reaction mixture was diluted with tetrahydrofuran (3 mL) and quenched with HCl aqueous solution (3 M, 1 mL). The resulting mixture was stirred at room temperature for 2 hours. The phases were then separated, the aqueous layer was extracted with ethyl acetate (5 mL × 3). The combined organic layer was washed with saturated NaHCO$_3$ solution and brine, dried over Na$_2$SO$_4$, filtered and concentrated *in vacuo*. The linear and branched ratio was determined by $^1$H NMR analysis of the crude reaction mixture. The residue was purified by column chromatography on silica gel (*n*-hexane: ethyl acetate = 10:1) to afford the desired product.

### General procedure for mechanochemical iron-catalyzed regiodivergent hydroarylation of aryl alkenes

In the glove box, a mixture of indole substrate **1** (0.2 mmol), aryl alkene **48** (0.3 mmol), Fe(acac)$_3$ (10 mol%, 0.02 mmol, 7.1 mg), **NHC preligand** (10 mol%, 0.02 mmol), magnesium chip (50 mol%, 0.1 mmol, 2.4 mg), TMEDA (0.2 mmol, 30 μL) and tetrahydrofuran (50 μL) were placed in a nitrogen-purged stainless-steel vessel (5 mL) with a stainless-steel ball ($d_{MB}$ = 7 mm). Then, the vessel was sealed and milled in a mixer mill (RETSCH MM 400) at 30 Hz for 270 min under a nitrogen atmosphere. Then, the reaction mixture was diluted with tetrahydrofuran (3 mL) and quenched with HCl aqueous solution (3 M, 1 mL). The resulting mixture was stirred at room temperature for 2 hours. The phases were then separated, the aqueous layer was extracted with ethyl acetate (5 mL × 3). The combined organic layer was washed with saturated NaHCO$_3$ solution and brine, dried over Na$_2$SO$_4$, filtered and concentrated *in vacuo*. The linear and branched ratio was determined by $^1$H NMR analysis of the crude reaction mixture. The residue was purified by column chromatography on silica gel (*n*-hexane: ethyl acetate = 5:1) to afford the desired product.

## Data availability

The data that support the findings of this study are available within the main text and the *Supplementary Information*. Details about materials, methods, experimental procedures, characterization data, NMR spectra and DFT-optimized structures are available in the *Supplementary Information*, all other data are available from the corresponding author upon request.

## Code availability

The Python codes for developing the MLVR model and LOO cross-validation are provided in *Supplementary Information*.

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

## Acknowledgements

The authors gratefully acknowledge the support from the ERC Advanced Grant (no. 101021358) and the DFG (Gottfried-Wilhelm-Leibniz-Preis and SPP2363) to L.A., the Alexander von Humboldt Foundation (fellowship to Z.-J. Z.), and the CSC scholarship (Z.L.).

## Author contributions

L.A. and Z.-J.Z. conceived the project. Z.-J. Z. designed the experiments and analyzed the data. Z.-J.Z. and Z.L. performed the experiments. X.C. performed the computational studies. All the authors participated in the discussion and preparation of the manuscript.

## Funding

## Competing interests

The authors declare no competing interests.
