## [Transparent Peer Review file · Nature Communications]

Mechanochemical ligand-controlled regiodivergent hydroarylation of alkenes via iron-catalyzed C–H activation

Corresponding Author: Professor Lutz Ackermann

Version 0:

Reviewer comments:

Reviewer #1

(Remarks to the Author)

Ackermann and co-workers report a mechanochemical iron-catalyzed anti-Markovnikov hydroarylations of unactivated alkenes using a bis(N-heterocyclic carbene) ligand, as well as regiodivergent hydroarylation of aryl alkenes by changing the N-heterocyclic carbene ligand. The products are generated in moderate to good yields with excellent selectivity is controlled by ligand. The coupling partners are unactivated alkenes and indoles, and the catalytic cycle proceeds via reductive turnover with magnesium metal as a convenient reductant to form the catalytically active iron(0) species. Under the standard reaction conditions, 75 target products were obtained in good to moderate yields. The methodology featured excellent regioselectivity, mild reaction conditions, and a broad substrate scope. Furthermore, the investigation of mechanism in the paper is well conducted, and an appropriate mechanistic proposal is given based on experimental results and DFT studies. Therefore, the manuscript may be publishable after a minor revision.

1. How about NH free indole in this system?
2. As shown in Fig. 5 the scope for ligand-controlled regiodivergent hydroarylation of aryl alkenes, only styrene was used as substrate, is it possible to use unactivated alkenes as starting material as shown in Fig. 3?
3. When 7-ethyl substituted indole was used, only 48% yield of 70 was obtained. The authors should give some comments on it.
4. Mg chip seems quite important for this reaction. What was the role of Mg chip in this transformation?

Reviewer #2

(Remarks to the Author)

Reviewer Report – Nature Communications

Summary

The manuscript by Zhang et al. describes a mechanochemical, ligand-controlled, regiodivergent hydroarylation of alkenes with NPMP-protected indoles via Fe/NHC-catalyzed C–H activation, using magnesium metal as the reductant. By selecting between mono- and bis(NHC) ligands, the authors achieve either Markovnikov or anti-Markovnikov selectivity, with application to unactivated alkenes, styrenes, and prefunctionalized alkene–complex molecule conjugates. Mechanistic insight is provided via DFT and multivariate linear regression (MVLN) analysis.

The concept of combining iron catalysis, ligand control, and mechanochemistry is interesting and timely. However, in its current form, the manuscript has several weaknesses that limit its impact and suitability for Nature Communications. These primarily relate to the justification and demonstration of mechanochemical benefits, substrate scope breadth, and the depth of mechanistic and comparative analysis.

Major Points

1. Justification for Mechanochemistry

While mechanochemistry is highlighted as a key innovation, there is no compelling experimental evidence that this mode of activation is essential for the reported transformation. The only control—failure of the reaction under thermal solution conditions at 100 °C—is insufficient. A direct comparison of mechanochemical vs. solution-phase reactions under otherwise matched conditions (e.g., Mg reduction in solvent, sonication, stirring) is needed to convincingly demonstrate any unique rate, selectivity, or functional group tolerance benefits.

2. Lack of Optimization of Mechanochemical Parameters

No systematic optimization of milling variables (frequency, time, ball size/material, ball-to-powder ratio) is reported. Given

the sensitivity of mechanochemical efficiency to these parameters, such optimization is critical to reproducibility, mechanistic understanding, and scale-up.

3. Narrow Substrate Class – Indoles Only

Although framed as a ligand-controlled, regiodivergent hydroarylation via C–H activation, the scope is almost exclusively limited to NPMP-indoles (and two azaindole). This narrows the significance of the concept. Demonstrating the same regiodivergent control with other heteroarenes (e.g., benzofurans, benzothiophenes, pyrroles, quinolines) would substantially strengthen the generality and impact.

4. Choice of Directing Group (NPMP)

The rationale for selecting N-para-methoxyphenyl (NPMP) as the directing group is not provided, and no data are shown for alternative directing groups. It is unclear if NPMP is uniquely effective, or if the methodology is more general. Testing common DGs such as picolinamides, carbamates, or sulfonamides would help assess the true scope.

5. Late-Stage Diversification Claims

The claim of "late-stage diversification of complex molecules" is overstated. The showcased examples are alkene–complex molecule conjugates prepared prior to the hydroarylation step, not native bioactive molecules containing alkenes. Demonstration on natural alkene-containing drugs or metabolites would make this claim credible.

6. Missing C4 and C7 Indole Substrates

Since steric and electronic effects can strongly influence C–H activation, inclusion of C4- and C7-substituted indoles would help define the method's limitations.

7. Regioselectivity Rationalization

The MVLR analysis is a valuable mechanistic addition but is based on a small ligand dataset and achieves only moderate predictive power ($R^2 = 0.72$). The practical applicability of the model should be discussed, ideally with external validation on additional ligands.

Additional Points

- Comparative Analysis – A more explicit benchmarking against recent solution-phase Fe- and Co-catalyzed hydroarylations of indoles and other heteroarenes is warranted, especially regarding selectivity, yield, cost, and environmental footprint.
- Mechanistic Evidence – The mechanistic proposal is plausible, but kinetic isotope effect (KIE) data, radical trapping experiments, and tests to rule out single-electron pathways are missing. Given the reductive Mg conditions, the involvement of SET should be addressed.

Recommendation

While the work is conceptually attractive, the current scope, mechanistic justification, and mechanochemistry validation are insufficient for Nature Communications. I recommend rejection, with submission to Communications Chemistry once the following concerns are addressed.

- (i) Provide clear experimental evidence for the necessity and advantage of mechanochemistry over conventional methods.
- (ii) Broaden the substrate scope to at least one additional heteroarene class and
- (iii) Perform the additional control experiments

Reviewer #3

(Remarks to the Author)

In this Ackerman and coworkers describe mechanochemical Fe/NHC catalyzed hydroalkylation of indols. Depending on either monodentate NHC or bidentate NHC ligand, the reaction shows interesting and highly appealing form of regiodivergency with respect to alkene reagent. The work, naturally stems from previous work on Fe-catalyzed C-H alkylation by authors teams and other groups, but the work goes very much beyond often quoted but I would say misunderstood "only mechanochemical application of known solution synthesis". This work bring several highly beneficial advancements connected with the use of mechanochemistry. Most notably, direct use and in situ activation of metallic magnesium rather than pre-formation of Grignard reagents. Same goes for highly unstable low valent iron complexes. So, mechanochemistry bring significant advance to this kind of C-H activation process. Moreover, authors studied mechanistic details of this transformation by some well-selected cross-experiments and DFT calculations. Overall, the study brings significant new knowledge to the catalytic synthesis under mechanical conditions showing that metal-catalyzed reactions are well compatible and can draw significant benefits from mechanochemical activation. The comments from previous round of reviewing were satisfactorily answered. Overall, the study was very competently performed and I recommend its acceptance by Nat. Commun. after below mentioned comments are considered by authors.

1) For DFT calculations rather standard method was used with vacuum based geometries and THF solvation for energies due to THF used as a LAG. There is very little information in the present literature on how should mechanochemical processes be modeled with respect to medium. Typically LAGs would still be used in large excess so I think this approach may still be the best compromise, but a comment with this respect would be useful to readers.

2) Some comment on the optimization of mechanical parameters would be useful, e.g. frequency, time, number of balls etc.

3) Important question with respect to Fe-catalyzed process and the use of stainless steel milling jars and especially balls is the possible contamination of the reaction with secondary scratched iron which can happen with the present abrasive materials in the system. Although optimization experiments without Fe-catalyst suggest that any background reaction if present is very slow, typical way to check this is to use some other material for balls and jars if they are available to authors.

Version 1:

Reviewer comments:

Reviewer #1

(Remarks to the Author)

In their revised version, Ackermann and co-workers answered the concern of the reviewers and have now an excellent work of high interest in the field of hydroarylation of alkenes. The authors have answered all of the questions by the reviewers and adopted the suggestions. The synthetically important paper can now be published immediately as it is.

(Remarks on code availability)

Reviewer #2

(Remarks to the Author)

In the revised manuscript entitled Mechanochemical ligand-controlled regiodivergent hydroarylation of alkenes via iron-catalyzed C–H activation, the authors have tried their best to address the concerns raised by the reviewer. I suggest that the manuscript can be accepted at this present form.

(Remarks on code availability)

Reviewer #3

(Remarks to the Author)

In this revised manuscript, authors addressed all my concerns and reflected on all suggestion in satisfactory way. As far as I can they, also points raised by other reviewers were also solved. Therefore, I recommend acceptance of this work by Nature Communications.

(Remarks on code availability)

Reply to comments by Reviewer 1

We appreciate **Reviewer 1** for favorable comments and many helpful suggestions!

Comments: Ackermann and co-workers report a mechanochemical iron-catalyzed anti-Markovnikov hydroarylations of unactivated alkenes using a bis(N-heterocyclic carbene) ligand, as well as regiodivergent hydroarylation of aryl alkenes by changing the N-heterocyclic carbene ligand. The products are generated in moderate to good yields with excellent selectivity is controlled by ligand. The coupling partners are unactivated alkenes and indoles, and the catalytic cycle proceeds via reductive turnover with magnesium metal as a convenient reductant to form the catalytically active iron(0) species. Under the standard reaction conditions, 75 target products were obtained in good to moderate yields. The methodology featured excellent regioselectivity, mild reaction conditions, and a broad substrate scope. Furthermore, the investigation of mechanism in the paper is well conducted, and an appropriate mechanistic proposal is given based on experimental results and DFT studies. Therefore, the manuscript may be publishable after a minor revision.

Question 1: How about NH free indole in this system?

Answer: We are grateful for the valuable suggestion from the reviewer. We performed the reaction with NH free indole. However, the target product was not obtained, and 84% of the indole starting material was recovered.

Question 2: As shown in Fig. 5 the scope for ligand-controlled regiodivergent hydroarylation of aryl alkenes, only styrene was used as substrate, is it possible to use unactivated alkenes as starting material as shown in Fig. 3?

Answer: We appreciate the valuable suggestion from the reviewer. Similar to the previous reports (*Org. Lett.* **2015**, *17*, 442; *Angew. Chem. Int. Ed.* **2017**, *56*, 14197; *Nat. Commun.* **2024**, *15*, 3503), the use of monodentate NHC ligand only tolerated aryl alkenes, while unactivated alkenes failed to afford the desired Markovnikov-selective products. In contrast, due to the unique structure and strong electron-donating ability of bidentate bis(NHC) ligand, both unactivated and aryl alkenes gave the desired *anti*-Markovnikov-selective products.

Question 3: When 7-ethyl substituted indole was used, only 48% yield of **70** was obtained. The authors should give some comments on it.

Answer: We thank the reviewer for this comment. The observed low reactivity at with 7-substituted indoles may be attributed to the steric hindrance induced by the substituent at this position, which may strongly affect the bond formation at the 2-position, as in compound **70** (48% yield). Similarly, when the indoles bear a sterically hindered *N*-substituent, the reaction efficacy was shown to decrease, such as in **6** (47% yield) and **7** (54% yield). We have included this comment in the revised manuscript as follows: “While, the substituent at the 7-position of indole, which is spatially proximal to the 2-position, imposed steric hindrance that affects the bond formation at the 2-position (**70**).”

Question 4: Mg chip seems quite important for this reaction. What was the role of Mg chip in this transformation?

Answer: We thank the reviewer for this question. Based on our control experiments (**Fig. 7a**) and previous relevant literature reports (*Adv. Synth. Catal.* **2016**, 358, 2564; *Nat. Commun.* **2024**, 15, 3503; *J. Am. Chem. Soc.* **2025**, 147, 6897), we believe that magnesium play a major role in reducing high-valent iron to zero-valent iron species under mechanochemical conditions. This avoids the use of Grignard reagents, and is indicative that it does not participate in bond activation or formation processes.

Reply to comments by Reviewer 2

We appreciate **Reviewer 2** for favorable comments and many helpful suggestions!

Comments: The manuscript by Zhang et al. describes a mechanochemical, ligand-controlled, regiodivergent hydroarylation of alkenes with NPMP-protected indoles via Fe/NHC-catalyzed C–H activation, using magnesium metal as the reductant. By selecting between mono- and bis(NHC) ligands, the authors achieve either Markovnikov or anti-Markovnikov selectivity, with application to unactivated alkenes, styrenes, and prefunctionalized alkene – complex molecule conjugates. Mechanistic insight is provided via DFT and multivariate linear regression (MVLRL) analysis.

The concept of combining iron catalysis, ligand control, and mechanochemistry is interesting and timely. However, in its current form, the manuscript has several weaknesses that limit its impact and suitability for Nature Communications. These primarily relate to the justification and demonstration of mechanochemical benefits, substrate scope breadth, and the depth of mechanistic and comparative analysis.

Question 1: Justification for Mechanochemistry

While mechanochemistry is highlighted as a key innovation, there is no compelling experimental evidence that this mode of activation is essential for the reported transformation. The only control – failure of the reaction under thermal solution conditions at 100 °C – is insufficient. A direct comparison of mechanochemical vs. solution-phase reactions under otherwise matched conditions (e.g., Mg reduction in solvent, sonication, stirring) is needed to convincingly demonstrate any unique rate, selectivity, or functional group tolerance benefits.

Answer: We thank the reviewer for this comment. As suggested, we have performed the reaction under solvent conditions (**Fig. 2; L10**), in which an equivalent amount of magnesium chip was added with the temperature raised to 100 °C, and strong stirring. However, no product formation was observed, which highlighted the necessity of mechanochemistry as a driving force over solvent conditions. Moreover, the use of Grignard reagents can be avoided under mechanochemistry, which greatly improves the functional groups compatibility, such as cyano and ester substituents (**Fig. 3, products 12 and 13**).

Question 2: Lack of Optimization of Mechanochemical Parameters

No systematic optimization of milling variables (frequency, time, ball size/material, ball-to-powder ratio) is reported. Given the sensitivity of mechanochemical efficiency

to these parameters, such optimization is critical to reproducibility, mechanistic understanding, and scale-up.

Answer: We are grateful for this valuable suggestion. As suggested, we have performed detailed optimization of the mechanochemical parameters and added these new results to the modified *Supplementary Information (Table S3)*.

Entry	Milling Time (min)	Milling Frequency (Hz)	Milling Ball (No.)	Yield (%)
1	180	30	7 mm (1)	92 (90)
2	90	30	7 mm (1)	31
3	270	30	7 mm (1)	91
4	180	25	7 mm (1)	74
5	180	20	7 mm (1)	65
6	180	30	10 mm (1)	89
7	180	30	3 mm (20)	85
8	180	30	3 mm (10)	64

Reaction conditions: **1a** (0.1 mmol), **2a** (0.15 mmol), Fe(acac)₃ (10 mol%), **L10** (10 mol%), Mg chip (50 mol%), TMEDA (0.1 mmol) and THF (50 µL) were placed in a stainless-steel vessel (5 mL) with stainless-steel ball(s), milled in a mixer mill (RETSCH MM 400) at x Hz for t min under nitrogen atmosphere. Then THF (3 mL) and HCl aq. (3 M, 1 mL) were added and the mixture was stirred for 2 h. The ratio of l:b is >99:1 for all cases. The yield was determined by ¹H NMR spectroscopy using 1,3,5-trimethoxybenzene as the internal standard (the yield of the isolated product is given within parentheses).

Question 3: Narrow Substrate Class – Indoles Only

Although framed as a ligand-controlled, regiodivergent hydroarylation via C–H activation, the scope is almost exclusively limited to NPMP-indoles (and two azaindole). This narrows the significance of the concept. Demonstrating the same regiodivergent control with other heteroarenes (e.g., benzofurans, benzothiophenes, pyrroles, quinolines) would substantially strengthen the generality and impact.

Answer: We sincerely appreciate the comment from the reviewer. We attempted the C–H alkylation of the benzene ring with different directing groups under mechanochemistry conditions, but the target products were not obtained. In addition, we tried the C–H alkylation of pyrrole, triazole, benzo[*d*]oxazole and benzofuran under the same mechanochemistry conditions, which were also unsuccessful (**Fig. R1**).

Fig. R1. Test reactions with different arenes

Question 4: Choice of Directing Group (NPMP)

The rationale for selecting N-para-methoxyphenyl (NPMP) as the directing group is not provided, and no data are shown for alternative directing groups. It is unclear if NPMP is uniquely effective, or if the methodology is more general. Testing common DGs such as picolinamides, carbamates, or sulfonamides would help assess the true scope.

Answer: We thank the reviewer for the valuable suggestion. The reaction does not occur in the absence of a directing group or using aldehyde, amide, imide, sulfonamide as the directing group. Using a phenyl-substituted imine directing group can achieve a yield of 53%, while introducing a methyl group at the *ortho* position or a trifluoromethyl group at the *para* position will decrease the reaction efficiency (**Fig. R2**). Notably, the functional formyl group in the product can undergo various late-stage transformations (**Fig. 6**) and can also be removed under palladium-catalyzed condition (**79**), facilitating the further modification of the products.

Fig. R2. Test reactions with different directing groups

Question 5: Late-Stage Diversification Claims

The claim of “late-stage diversification of complex molecules” is overstated. The showcased examples are alkene–complex molecule conjugates prepared prior to the hydroarylation step, not native bioactive molecules containing alkenes. Demonstration on natural alkene-containing drugs or metabolites would make this claim credible.

Answer: We thank the reviewer’s comment on late-stage diversification. The diversification reported in **Fig. 6** showcases the broad scope of transformation starting from the hydroarylation product. Through conversion of the formyl group and directed C4–H activation, various functional groups can be introduced into the 3- and 4-positions of the indole ring. This method enables the selective functionalization of multiple sites on the indole ring of a common synthetic building block, which has important implications for the synthesis and subsequent modification of bioactive molecules.

Question 6: Missing C4 and C7 Indole Substrates

Since steric and electronic effects can strongly influence C–H activation, inclusion of C4- and C7-substituted indoles would help define the method’s limitations.

Answer: We thank the reviewer for this comment. As suggested by the reviewer, we have introduced electron-withdrawing as well as electron-donating substituents at the 4- and 7-positions of indole substrates, where good yields and regioselectivities could be achieved (**Fig. 3; 9,18,19**).

Question 7: Regioselectivity Rationalization

The MVLr analysis is a valuable mechanistic addition but is based on a small ligand dataset and achieves only moderate predictive power ($R^2 = 0.72$). The practical applicability of the model should be discussed, ideally with external validation on additional ligands.

Answer: To test the applicability of the MVLr model, we performed leave-one-out (LOO) cross-validation given the small scale of the available dataset, where each sample was used once as the singleton validation set while the remaining sample formed the training set. The predictive performance of MVLr model was moderate with an R^2 value of 0.71 after LOO cross-validation, but it still underscored the relationship between the chosen parameters of the key triplet iron(0)-NHC complex and the regioselectivity, including cone angle, buried volume and spin density exemplified in **Fig. R3**. The model performance in LOO cross-validation has been included in the revised manuscript and the Python script for building the MVLr model in the *Supplementary Information*.

Fig. R3. Multivariate linear regression model using four parameters derived from key triplet iron(0) complex.

Question 8: Comparative Analysis – A more explicit benchmarking against recent solution-phase Fe- and Co-catalyzed hydroarylations of indoles and other heteroarenes

is warranted, especially regarding selectivity, yield, cost, and environmental footprint.

Mechanistic Evidence – The mechanistic proposal is plausible, but kinetic isotope effect (KIE) data, radical trapping experiments, and tests to rule out single-electron pathways are missing. Given the reductive Mg conditions, the involvement of SET should be addressed.

While the work is conceptually attractive, the current scope, mechanistic justification, and mechanochemistry validation are insufficient for Nature Communications. I recommend rejection, with submission to Communications Chemistry once the following concerns are addressed.

Answer: We thank the reviewer for these suggestions. This reaction achieves excellent yields and regioselectivity through mechanochemical iron-catalyzed hydroarylation. Compared to solvent-phase reactions, it offers numerous advantages, including milder reaction conditions, minimal solvent usage, and the avoidance of Grignard reagents, which enhance functional group compatibility. Furthermore, this reaction represents the first use of a bis(NHC) ligand in 3d metal-catalyzed hydroarylation, achieving excellent *anti*-Markovnikov selectivity.

The KIE experiments were provided in *Supplementary Information (Fig. S1-S3)*.

Kinetic isotope effect experiments

Entry	Time (min)	Yield (%) (with 1a)	Yield (%) (with $[D]_1\text{-1a}$)
1	2	0.5	0.5
2	4	4.5	4
3	6	8	7
4	8	11	10
5	10	14	12.5
6	12	17.5	15

Entry	Time (min)	Yield (%) (with 1a)	Yield (%) (with $[D]_1$ - 1a)
1	5	2	0.5
2	10	5	3
3	15	8.5	7
4	20	14	10.5
5	25	18.5	14.5
6	30	23	16.5

Entry	Time (min)	Yield (%) (with 1a)	Yield (%) (with $[D]_1\text{-1a}$)
1	20	1.5	0.5
2	40	4	2.5
3	60	6.5	3
4	80	9	6
5	100	13	9
6	120	15	11

Based on our control experiments (**Fig. 7a**) and previous relevant literature reports (*Adv. Synth. Catal.* **2016**, 358, 2564; *Nat. Commun.* **2024**, 15, 3503; *J. Am. Chem. Soc.* **2025**, 147, 6897), we believe that the role of magnesium is reducing high-valent iron to zero-valent iron species under mechanochemical conditions, avoiding the use of Grignard reagents, and it does not participate in bond activation or formation processes.

We also performed radical trapping experiments (**Fig. R4**). Adding TEMPO or BHT to the reaction system will terminate the reaction. A large amount of substrate **1a** can be recovered, but no by-products of the free radical terminators binding with substrate were found. We believe that free radical terminators affect the electron transfer process of magnesium-reduced iron catalysts, which hinders the smooth progress of the reaction.

Fig. R4. Radical trapping experiments

Question 9: Provide clear experimental evidence for the necessity and advantage of mechanochemistry over conventional methods.

Answer: We thank the reviewer for this comment. As suggested, we have performed the reaction under solvent conditions (**Fig. 2; L10**), in which an equivalent amount of magnesium chip was added with the temperature raised to 100 °C, and strong stirring. However, no product formation was observed, which highlighted the necessity of mechanochemistry as a driving force over solvent conditions. Moreover, the use of Grignard reagents can be avoided under mechanochemistry, which greatly improves the functional group compatibility, such as cyano and ester substituents (**Fig. 3**, products **12** and **13**).

Question 10: Broaden the substrate scope to at least one additional heteroarene class.

Answer: Taking the reviewer's comment into consideration, we applied our strategy to C–H alkylation of the benzene ring with a wide variety of directing groups under mechanochemistry conditions. However, under otherwise reaction conditions, these gave thus far unfruitful results **Fig. R1**). Additionally, pyrrole, triazole, benzo[*d*]oxazole, and benzofuran were also considered under the same mechanochemistry conditions, but were deemed unsuitable substrates for the desired transformation (**Fig. R1**).

Fig. R1. Test reactions with different arenes

Question 11: Perform the additional control experiments.

Answer: We are grateful for the valuable suggestion from the reviewer. We have performed detailed optimization of the mechanochemical parameters and added the results to the *Supplementary Information* (**Table S3**).

Entry	Milling Time (min)	Milling Frequency (Hz)	Milling Ball (No.)	Yield (%)
1	180	30	7 mm (1)	92 (90)
2	90	30	7 mm (1)	31
3	270	30	7 mm (1)	91
4	180	25	7 mm (1)	74
5	180	20	7 mm (1)	65
6	180	30	10 mm (1)	89
7	180	30	3 mm (20)	85
8	180	30	3 mm (10)	64

Reaction conditions: **1a** (0.1 mmol), **2a** (0.15 mmol), Fe(acac)₃ (10 mol%), **L10** (10 mol%), Mg chip (50 mol%), TMEDA (0.1 mmol) and THF (50 μL) were placed in a stainless-steel vessel (5 mL) with stainless-steel ball(s), milled in a mixer mill (RETSCH MM 400) at x Hz for t min under nitrogen atmosphere. Then THF (3 mL) and HCl aq. (3 M, 1 mL) were added and the mixture was stirred for 2 h. The ratio of l:b is >99:1 for all cases. The yield was determined by ¹H NMR spectroscopy using 1,3,5-trimethoxybenzene as the internal standard (the yield of the isolated product is given within parentheses).

The KIE experiments were provided in *Supplementary Information* (**Fig. S1-S3**).

Reply to comments by Reviewer 3

We appreciate **Reviewer 3** for favorable comments and many helpful suggestions!

Comments: In this Ackermann and coworkers describe mechanochemical Fe/NHC catalyzed hydroalkylation of indols. Depending on either monodentate NHC or bidentate NHC ligand, the reaction shows interesting and highly appealing form of regiodivergency with respect to alkene reagent. The work, naturally stems from previous work on Fe-catalyzed C-H alkylation by authors teams and other groups, but the work goes very much beyond often quoted but I would say misunderstood "only mechanochemical application of known solution synthesis". This work bring several highly beneficial advancements connected with the use of mechanochemistry. Most notably, direct use and in situ activation of metallic magnesium rather than pre-formation of Grignard reagents. Same goes for highly unstable low valent iron complexes. So, mechanochemistry bring significant advance to this kind of C-H activation process. Moreover, authors studied mechanistic details of this transformation by some well-selected cross-experiments and DFT calculations. Overall, the study brings significant new knowledge to the catalytic synthesis under mechanical conditions showing that metal-catalyzed reactions are well compatible and can draw significant benefits from mechanochemical activation. The comments from previous round of reviewing were satisfactorily answered. Overall, the study was very competently performed and I recommend its acceptance by Nat. Commun. after below mentioned comments are considered by authors.

Question 1: For DFT calculations rather standard method was used with vacuum based geometries and THF solvation for energies due to THF used as a LAG. There is very little information in the present literature on how should mechanochemical processes be modeled with respect to medium. Typically LAGs would still be used in large excess so I think this approach may still be the best compromise, but a comment with this respect would be useful to readers.

Answer: We fully agree with reviewer's comments on the DFT computational methods for the liquid-assisted grinding reaction. As the effect of the continuum dielectric environment for our reaction cannot be neglected (**Table S2**), we included the solvent effect in single-point energy calculation after gas-phase geometry optimization. The same level of theory was proven to successfully reproduce the experimental results of the mechanochemical Diels-Alder reaction and the synthesis of sulfonylguanidines

(*ChemSusChem* **2021**, *14*, 2763–2768). This discussion about the computational methods has been added to the revised manuscript, as follows: “It is worth noticing that the solvent effect was included in the free energy calculation based on gas-phase optimized geometries, given that our reaction is liquid assisted grinding with THF⁶⁹.”, where the above reference has been cited.

Question 2: Some comment on the optimization of mechanical parameters would be useful, e.g. frequency, time, number of balls etc.

Answer: We are grateful for the valuable suggestion from the reviewer. We have performed detailed optimization of the mechanochemical parameters and added the results to the *Supplementary Information (Table S3)*.

Entry	Milling Time (min)	Milling Frequency (Hz)	Milling Ball (No.)	Yield (%)
1	180	30	7 mm (1)	92 (90)
2	90	30	7 mm (1)	31
3	270	30	7 mm (1)	91
4	180	25	7 mm (1)	74
5	180	20	7 mm (1)	65
6	180	30	10 mm (1)	89
7	180	30	3 mm (20)	85
8	180	30	3 mm (10)	64

Reaction conditions: **1a** (0.1 mmol), **2a** (0.15 mmol), Fe(acac)₃ (10 mol%), **L10** (10 mol%), Mg chip (50 mol%), TMEDA (0.1 mmol) and THF (50 μL) were placed in a stainless-steel vessel (5 mL) with stainless-steel ball(s), milled in a mixer mill (RETSCH MM 400) at x Hz for t min under nitrogen atmosphere. Then THF (3 mL) and HCl aq. (3 M, 1 mL) were added and the mixture was stirred for 2 h. The ratio of l:b is >99:1 for all cases. The yield was determined by ¹H NMR spectroscopy using 1,3,5-trimethoxybenzene as the internal standard (the yield of the isolated product is given within parentheses).

Question 3: Important question with respect to Fe-catalyzed process and the use of stainless steel milling jars and especially balls is the possible contamination of the reaction with secondary scratched iron which can happen with the present abrasive materials in the system. Although optimization experiments without Fe-catalyst suggest

that any background reaction if present is very slow, typical way to check this is to use some other material for balls and jars if they are available to authors.

Answer: We are grateful for this valuable suggestion. On the scale-up experiment (4 mmol scale), a zirconia vessel in combination with a zirconia ball was employed, delivering the desired product with excellent yield and regioselectivity (86% yield, l:b >99:1). This demonstrates the robustness of our strategy. Please see Fig. 6 and Supplementary Information (Scale-up reaction).